# Comparative Evaluation of a Standard M10 Assay with Xpert Xpress for the Rapid Molecular Diagnosis of SARS-CoV-2, Influenza A/B Virus, and Respiratory Syncytial Virus

**DOI:** 10.3390/diagnostics13233507

**Published:** 2023-11-22

**Authors:** Azwani Abdullah, I-Ching Sam, Yin Jie Ong, Chun Hao Theo, Muhammad Harith Pukhari, Yoke Fun Chan

**Affiliations:** 1Department of Medical Microbiology, Faculty of Medicine, Universiti Malaya, Kuala Lumpur 50603, Malaysia; jicsam@ummc.edu.my (I.-C.S.); jessoyj@um.edu.my (Y.J.O.); theoch723@gmail.com (C.H.T.); mhbpbs@gmail.com (M.H.P.); 2Department of Medical Microbiology, Universiti Malaya Medical Centre, Kuala Lumpur 59100, Malaysia

**Keywords:** respiratory syncytial virus, influenza A virus, influenza B virus, SARS-CoV-2, rapid RT-PCR, diagnostic accuracy, point-of-care testing, Malaysia

## Abstract

SARS-CoV-2, influenza A/B virus (IAV/IBV), and respiratory syncytial virus (RSV) are among the common viruses causing acute respiratory infections. Clinical diagnosis to differentiate these viruses is challenging due to similar clinical presentations; thus, laboratory-based real-time RT PCR is the gold standard for diagnosis. This retrospective study aimed to evaluate the diagnostic performance of STANDARD M10 Flu/RSV/SARS-CoV-2 (SD Biosensor Inc., Seoul, Korea) using archived positive and negative respiratory samples for SARS-CoV-2, IAV, IBV, and RSV. A total of 322 respiratory samples were tested, comprising 215 positive samples (49 SARS-CoV-2, 48 IAV, 53 IBV, 65 RSV) and 107 negative samples. All samples were tested with both STANDARD M10 and compared to either Xpert Xpress SARS-CoV-2 or Xpert Xpress Flu/RSV (Cepheid, Sunnyvale, CA, USA). The sensitivity, specificity, positive predictive value, and negative predictive value rates of STANDARD M10 were very similar to Xpert Xpress SARS-CoV-2 or Xpert Xpress Flu/RSV ranges for each virus (98–100%). The duration of testing and workflows were similar. The overall agreement was 99.4%, including 99.1% agreement for positive samples and 100% agreement for negative samples. In conclusion, the STANDARD M10 point-of-care test is suitable for rapid simultaneous detection of SARS-CoV-2, IAV, IBV, and RSV.

## 1. Introduction

Acute respiratory infections result in significant morbidity and mortality worldwide and are commonly caused by viruses including influenza viruses, parainfluenza viruses, respiratory syncytial viruses (RSVs), human rhinoviruses, human metapneumoviruses, and coronaviruses [1]. Following the relaxation of the infection control measures implemented during the COVID-19 period, an increase in transmission of these respiratory viruses has been reported [2]. In equatorial Malaysia, with COVID-19 remaining endemic, there is a year-round incidence of influenza [3] with no distinct seasonality, unlike countries in temperate regions. Patients may present some of these symptoms including cough, runny nose, sore throat, breathlessness, wheezing, headache, and fever. Differentiating these respiratory viruses solely based on clinical presentation is challenging due to their similar, non-specific signs and symptoms. With the recent COVID-19 pandemic, the need for rapid and robust diagnostic tools has gained more attention, not only for the institution of appropriate treatment but also for timely infection control measures and perhaps an early signal to detect a future potential outbreak [4].

The current diagnostic tools for detecting common respiratory viruses include a reverse transcription–polymerase chain reaction (RT-PCR) assay, virus culture, antigen detection, and antibody detection [5]. Real-time RT-PCR tests are the gold standard but may need several steps, including extraction and amplification, and require trained staff and costly laboratory set-up [6]. The diagnostic usefulness of virus culture is limited as the test is not widely available and may take up to 10–14 days. Although antigen and antibody detection are cheaper with a shorter turn-around time, sensitivity and specificity are lower compared to real-time RT-PCR, depending on the viral load of the specimen for antigen detection and antibody response of the infected patients [5]. Thus, a simple, rapid, and robust assay with diagnostic performance comparable to real-time RT-PCR is needed to overcome these diagnostic limitations. Differentiating these respiratory viruses is crucial not only to facilitate timely optimum treatment but also for the institution of infection control measures. With increasing demands in the market, several manufacturers have taken steps in developing rapid detection and point-of-care test kits for the simultaneous detection of influenza A virus (IAV), influenza B virus (IBV), RSV, and SARS-CoV-2 virus [7].

The STANDARD M10 Flu/RSV/SARS-CoV-2 (SD Biosensor Inc, Seoul, Korea) is a fully automated, scalable, rapid assay that enables simultaneous differentiation and detection of IAV, IBV, RSV, and SARS-CoV-2 virus from a single specimen. This assay uses a single device with results available in one hour [7]. Sample processing, nucleic acid extraction, and amplification take place in a fully automated, closed system, requiring minimal training and monitoring with less cross-contamination risk.

The objective of this study is to evaluate the diagnostic performance of the STANDARD M10 Flu/RSV/SARS-CoV-2 assay in comparison to Xpert Xpress SARS-CoV-2 or Xpert Xpress Flu/RSV (Cepheid, Sunnyvale, CA, USA) as a comparator test.

## 2. Materials and Methods

### 2.1. Sample Collection and Study Design

This retrospective study was conducted using residual respiratory samples from patients with respiratory infections sent to the diagnostic virology unit of Universiti Malaya Medical Centre (UMMC), Kuala Lumpur, Malaysia. The samples were stored at −80 °C and retrieved for this study.

At least 45 positive samples for each virus (IAV, IBV, RSV, and SARS-CoV-2) and 100 negative samples were collected. To achieve the desired sample sizes, positive respiratory samples with IAV, IBV, or RSV from May 2013 to January 2023, and positive samples for SARS-CoV-2 from February 2022 to January 2023 were retrieved. These samples were confirmed using a composite gold standard of multiple tests, such as immunofluorescence (IF), virus culture, or PCR, with at least one of the tests reported positive.

The samples were tested with both STANDARD M10 and either Xpert Xpress SARS-CoV-2 or Xpert Xpress Flu/RSV as the index test and comparator test, respectively. In brief, once thawed to room temperature, 300 µL of the sample was loaded into each STANDARD M10 and Xpert Xpress cartridge and set up for the automated PCR concurrently. Xpert Xpress Flu/RSV was used if the samples were previously reported as positive for IAV, IBV, or RSV, whereas Xpert Xpress SARS-CoV-2 was used if the samples were previously reported as positive for SARS-CoV-2.

The sensitivity, specificity, positive predictive value (PPV), negative predictive value (NPV), accuracy, and agreement rates were calculated.

### 2.2. Index Test

STANDARD M10 is an all-in-one cartridge-based ready-to-use multiplex RT-PCR assay intended for the qualitative detection of IAV, IBV, RSV, and SARS-CoV-2 RNA. It simultaneously detects the M gene for IAV, the NS1 gene for IBV, the M gene for RSV, and two gene targets (ORF1ab and N) for SARS-CoV-2. For SARS-CoV-2, the detection of at least one target signifies SARS-CoV-2 infection [7]. Once the sample was vortexed for about 10 s, 300 µL was added directly to the cartridge for RT-PCR in the STANDARD M10 analyzer according to the manufacturer’s instructions. The limit of detection (LoD) of this assay is 200 copies/mL for the SARS-CoV-2 ORF1ab gene, 400 copies/mL for IAV/IBV, and 800 copies/mL for both RSV and SARS-CoV-2 N genes [7].

### 2.3. Comparator Test

The Xpert Xpress SARS-CoV-2 and Xpert Xpress Flu/RSV assays are automated in vitro diagnostic tools performed on the GeneXpert Xpress Systems that carry out nucleic acid extraction, real-time PCR, and analysis. These tests are used for the qualitative detection and differentiation of various RNA viruses. Similar to STANDARD M10, 300 μL of the sample was loaded into the cartridge before PCR setup. The probes and primers used in Xpert Xpress SARS-CoV-2 amplify and detect the nucleocapsid (N2) and envelope (E) genes of the SARS-CoV-2 virus [8]. Xpert Xpress Flu/RSV detects the matrix (M) gene, basic polymerase (PB2), and acidic protein (PA) for IAV, the matrix (M) gene and non-structural protein (NS) for IBV, and nucleocapsid (N) for RSV [9].

### 2.4. Standard Tests

The samples used in this study were previously confirmed with various standard routine tests. SARS-CoV-2 was detected by real-time RT-qPCR SARS-CoV-2, while IAV, IBV, and RSV were detected by either an immunofluorescence assay or virus culture. IAV and IBV were also detected by multiplex RT-qPCR protocol 1 of the 2017 WHO guidelines with minor modifications [10]. A sample was considered positive for IAV, IBV, RSV, or SARS-CoV-2 if any one of these standard tests was positive.

#### 2.4.1. RT-qPCR for SARS-CoV-2

Either cobas SARS-CoV-2 (Roche, Basel, Switzerland) or Seegene Allplex SARS-CoV-2 assay (Seegene, Seoul, Korea) were used for the qualitative detection of SARS-CoV-2 RNA in respiratory samples. The coba SARS-CoV-2 assay is fully automated and detects the structural protein envelope (E) gene, which is specific for pan-Sarbecovirus and the ORF1 a/b, a non-structural region that is unique to SARS-CoV-2 [11]. For the Allplex SARS-CoV-2 assay, nucleic acid extraction was carried out separately with a GF-1 Nucleic Acid Extraction Kit (Vivantis Technologies, Shah Alam, Malaysia). Real-time PCR was performed on a CFX96 Real-Time PCR Detection System (Bio-Rad Laboratories, Hercules, CA, USA). The Allplex SARS-CoV-2 assay simultaneously detects four different target genes of SARS-CoV-2 (RdRP, S, and N genes specific for SARS-CoV-2 and the E gene for all Sarbecoviruses) [12]. For both assays, results were analyzed and interpreted using pre-installed software.

#### 2.4.2. Immunofluorescence Assay

The D3 Ultra 8 Direct Fluorescent Antibody Respiratory Virus Screening & Identification Kit (Diagnostic Hybrids, Athens, TN, USA) was performed on nasopharyngeal aspirates affixed to glass slides to detect a panel of respiratory viruses, including IAV, IBV, and RSV [13].

#### 2.4.3. Virus Culture

Samples were inoculated into HEp2, Vero, LLC-MK2, and MDCK cells and incubated at 37 °C with 5% CO_2_ for up to 10 days. Growth of the viruses was then detected by cytopathic effect, and respiratory viruses were identified using immunofluorescence.

### 2.5. Study Endpoint and Data Analysis

Our study endpoints were to evaluate the sensitivity, specificity, positive predictive value (PPV), negative predictive value (NPV), and accuracy rates of STANDARD M10 compared to Xpert Xpress SARS-CoV2 or Xpert Xpress Flu/RSV. The diagnostic validation parameters were calculated in percentages with 95% confidence intervals (CIs). The diagnostic agreement was also tested to measure the result concordance level. The Wilcoxon signed-rank test was used to analyze the Ct value distribution and the Spearman correlation was used for Ct value correlation. All data were statistically analyzed using GraphPad Prism version 9.0 (GraphPad Prism Software, Boston, MA, USA). The study findings were reported according to the Standards for Reporting of Diagnostic Accuracy Studies (STARD) statement (Figure 1). The use of residual diagnostic respiratory samples was approved by the UMMC medical ethics committee (No. 2022816-11476).

## 3. Results

### 3.1. Final Sample Size and Diagnostic Performance of STANDARD M10

A total of 322 respiratory samples were tested with both STANDARD M10 FLU/RSV/SARS-CoV-2 and either Xpert Xpress SARS-CoV-2 or Xpert Xpress Flu/RSV. There were 215 positive samples (48 IAV, 53 IBV, 65 RSV, and 49 SARS-CoV-2) and 107 negative samples (58 for IAV, IBV, and RSV and 49 for SARS-CoV-2) (Figure 1).
Figure 1The 322 samples included in this study with the results from the standard tests (orange boxes) and the test assay (blue boxes).
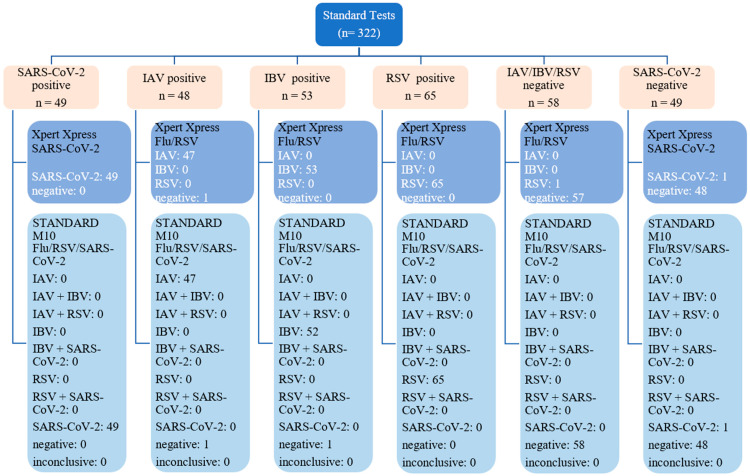


The overall diagnostic parameters (95% confidence interval) for STANDARD M10 and Xpert Xpress are shown in Table 1. For SARS-CoV-2, both STANDARD M10 and Xpert Xpress SARS-CoV-2 showed identical sensitivity, specificity, PPV, NPV, and accuracy, ranging between 98 and 100%. For IAV, STANDARD M10, and Xpert Xpress Flu/RSV had similar performances with a sensitivity of 97.9%, a specificity of 100%, a PPV of 100%, an NPV of 98.3%, and an accuracy of 99.1%. Non-significant differences were observed for IBV, with STANDARD M10 showing sensitivity, NPV, and accuracy rates of 98.1%, 98.3%, and 99.1%, respectively, compared to 100% for all diagnostic parameters of Xpert Xpress Flu/RSV. For RSV, STANDARD M10 showed 100% rates in all the diagnostic parameters compared to Xpert Xpress Flu/RSV, which showed 100% rates for sensitivity and NPV, 98.3% for specificity, 98.5% for PPV, and 99.2% for accuracy; these differences were not significant.

Our study had three inaccurate results each from STANDARD M10 and Xpert Xpress assays. STANDARD M10 showed one false positive result with SARS-CoV-2 (Ct 35.43) and one false negative result for both IAV and IBV. Xpert Xpress had one false positive result for both SARS-CoV-2 (Ct 41.80) and RSV (Ct 35.20) and one false negative result for IAV.

There was a close overall agreement of 99.4% (95% CI, 97.8–99.8%) between STANDARD M10 and Xpert Xpress, 99.1% (95% CI, 96.7–99.7) agreement for positive samples, and 100% (95% CI, 96.5–100.0) agreement for negative samples.

### 3.2. The Correlation between STANDARD M10 and Either Xpert Xpress SARS-CoV-2 or Xpert Xpress Flu/RSV

Both test assays showed very similar Ct value ranges in the positive samples for all the tested viruses, as shown in Table 2. The Ct values distribution and correlation (R^2^) comparing STANDARD M10 and Xpert Xpress SARS-CoV-2 or Xpert Xpress Flu/RSV are shown in Figure 2. Xpert Xpress recorded significantly higher Ct values for all viruses, with median cycle differences of 1.90 (IQR = 0.77) for SARS-CoV-2, 1.13 (IQR = 1.50) for IAV, 1.36 (IQR = 0.91) for IBV, and 1.76 (IQR = 3.47) for RSV. Overall, STANDARD M10 reported lower Ct values by two cycles compared to Xpert Xpress. For the Ct value correlation, SARS-CoV-2 had the highest R^2^ value of 0.98, followed by IBV at 0.93, IAV at 0.87, and RSV at 0.72 (all *p*-values < 0.0001).

## 4. Discussion

Standard tests used for the detection of respiratory viruses have many advantages but also limitations. In the past, cell culture was routinely used [13] but suffered a few limitations such as a longer turnaround time, being laborious, and carrying a high risk of cell contamination [6]. Compared to the culture method, immunofluorescence to detect antigens can provide a faster result for early clinical management and infection control measures but is generally limited due to its moderate sensitivity. These factors have swayed the preference for other diagnostic methods, such as rapid point-of-care PCR or rapid antigen tests, for the timely diagnosis of virus infection [6].

Since the COVID-19 pandemic, qPCR has gained wide usage. The cobas and Seegene assays offer turnaround times ranging from 2.5 to 3 h, while GeneXpert and STANDARD M10 offer shorter turnaround times of about an hour without compromising diagnostic performance. In addition to the faster result, the usage of multiplex PCR, such as the STANDARD M10 assay, increases the laboratory testing capacity and reduces the use of reagents. These increased throughputs can preserve laboratory and staff resources, making it a preferred diagnostic test.

Many studies have reported excellent performance of both Xpert Xpress SARS-CoV-2 [4,14,15,16] and Xpert Xpress Flu/RSV [15,16]. Our study shows that the recently introduced STANDARD M10 is comparable with these established assays, with overall sensitivity and specificity ranging from 98% to 100% for the detection of all four respiratory viruses.

STANDARD M10 has an advantage as it enables simultaneous detection of individual viruses from a single sample. As the clinical presentation of SARS-CoV-2 infection can be difficult to differentiate from influenza and RSV [6], multiple samplings may be needed for testing with the different Xpert Xpress assays. Multiple or repeated sampling not only contributes to patient discomfort but also a longer turnaround time and higher cost if both of the Xpert Xpress tests were to be performed separately. The ability of STANDARD M10 to simultaneously differentiate and detect IAV, IBV, RSV, and SARS-CoV-2 is the main advantage of this assay, especially for emergency use in settings with poor laboratory facilities. Another additional advantage of STANDARD M10 is scalability, with each console expandable to a maximum of eight modules. This can help the laboratory to scale up and down accordingly during the epidemic seasonality of the viruses.

The first reported evaluation study of STANDARD M10 in Italy showed an overall agreement as high as 94.6%, with positive percent agreement ranging from 96.6% to 100% and negative percent agreement ranging from 98.4% to 100% [7]. Our study with 322 samples shows consistent findings with this study, reflecting the comparable performance of STANDARD M10 in different regions of the world. Our study also used samples as early as 2022 for SARS-CoV-2, 2018 for IAV, and 2013 for IBV and RSV, suggesting that STANDARD M10 can detect a wide range of viruses that have evolved over time.

The pre-set cut-off Ct value is 40 for STANDARD M10 and 45 for Xpert Xpress. However, Ct values are an unreliable surrogate for determining viral load, infectivity, or transmissibility of the viruses [17]. The Ct values may be affected by nucleic acid extraction, primer designing, assay sensitivity (limit of detection), and the Ct value determination method of various assays [18]. Thus, these inter-test variabilities of Ct values between various rtRT-PCRs are not critical nor directly comparable. As reported in another study, Ct values are not recommended to be used as a guide for making clinical decisions [17].

This study, however, was not correlated with the patients’ epidemiology data and clinical presentations. The collected sample was anonymous and only labeled with a unique identification number for stock purposes. The sample sizes were not equally distributed among tested viruses, as sample availability may be affected by the seasonality of the specific virus.

## 5. Conclusions

In conclusion, STANDARD M10 demonstrated high sensitivity and specificity for the detection of IAV, IBV, RSV, and SARS-CoV-2 virus from clinical respiratory samples. The diagnostic performance of STANDARD M10 is comparable to the established Xpert Xpress SARS-CoV-2 or Xpert Xpress Flu/RSV assays. With the capabilities for simultaneous detection of these four important respiratory viruses from a single sample within an hour, in addition to the fully automated system, STANDARD M10 can be used as a point-of-care test assay in an emergency or critical care department or facilities without sophisticated laboratory set-ups. This assay is particularly useful in tropical countries, like Malaysia, which have year-round endemic influenza and COVID-19.

## Figures and Tables

**Figure 2 diagnostics-13-03507-f002:**
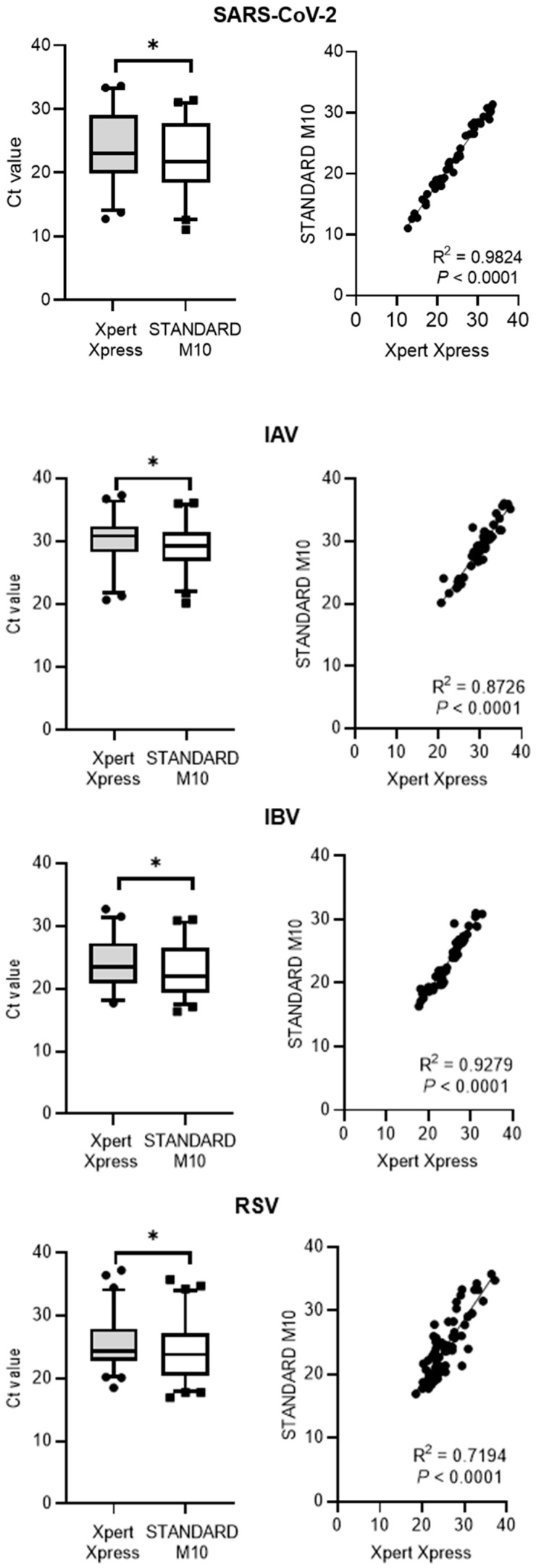
Ct values distribution and correlation comparing STANDARD M10 and either Xpert Xpress SARS-CoV-2 or Xpert Xpress Flu/RSV. Ct distribution is shown as medians with upper and lower quartiles, with outliers as individual points outside the whisker plot. An asterisk (*) denotes *p* < 0.001.

**Table 1 diagnostics-13-03507-t001:** Diagnostic parameters of STANDARD M10 and Xpert Xpress SARS-CoV-2 and Xpert Xpress Flu/RSV.

Virus/Parameters % (95% CI)	Xpert Xpress SARS-CoV-2/Xpert Xpress Flu/RSV	STANDARD M10
**SARS-CoV-2**		
Sensitivity	100.0 (92.8–100.0)	100.0 (92.8–100.0)
Specificity	98.0 (89.2–100.0)	98.0 (89.2–100.0)
PPV	98.0 (89.4–100.0)	98.0 (89.4–100.0)
NPV	100.0 (92.6–100.0)	100.0 (92.6–100.0)
Accuracy	99.0 (94.5–100.0)	99.0 (94.5–100.0)
**IAV**		
Sensitivity	97.9 (88.9–100.0)	97.9 (88.9–100.0)
Specificity	100.0 (93.8–100.0)	100.0 (93.8–100.0)
PPV	100.0 (92.5–100.0)	100.0 (92.5–100.0)
NPV	98.3 (90.9–100.0)	98.3 (90.9–100.0)
Accuracy	99.1 (94.9–100.0)	99.1 (94.9–100.0)
**IBV**		
Sensitivity	100.0 (93.3–100.0)	98.1 (89.9–100.0)
Specificity	100.0 (93.8–100.0)	100.0 (93.8–100.0)
PPV	100.0 (93.3–100.0)	100.0 (93.2–100.0)
NPV	100.0 (93.8–100.0)	98.3 (90.9–100.0)
Accuracy	100.0 (96.7–100.0)	99.1 (95.1–100.0)
**RSV**		
Sensitivity	100.0 (94.5–100.0)	100.0 (94.5–100.0)
Specificity	98.3 (90.8–100.0)	100.0 (93.8–100.0)
PPV	98.5 (91.8–100.0)	100.0 (94.5–100.0)
NPV	100.0 (93.7–100.0)	100.0 (93.8–100.0)
Accuracy	99.2 (95.6–100.0)	100.0 (97.1–100.0)

**Table 2 diagnostics-13-03507-t002:** Ct value ranges for all target genes by Xpert Xpress SARS-CoV-2 or Xpert Xpress Flu/RSV and STANDARD M10.

Test Assay	Virus	Target Gene	Ct Value Range (Median ± Interquartile Range)
**Xpert Xpress SARS-CoV-2 or Xpert Xpress Flu/RSV**	SARS-CoV-2	E	12.10–32.40 (22.00 ± 8.30)
		N2	13.40–34.90 (24.30 ± 9.20)
	IAV	A1	19.80–36.70 (30.00 ± 4.05)
		A2	21.50–38.50 (31.60 ± 3.40)
	IBV	IBV	17.70–32.70 (23.60 ± 6.10)
	RSV	RSV	18.50–37.20 (24.30 ± 4.80)
**STANDARD M10**	SARS-CoV-2	ORF1ab	11.60–31.02 (22.01 ± 8.56)
		N	10.52–31.77 (21.3 ± 9.22)
	IAV	IAV	20.14–35.95 (29.26 ± 4.25)
	IBV	IBV	16.35–30.99 (21.93 ± 7.07)
	RSV	RSV	16.92–35.71 (23.77 ± 6.24)

## Data Availability

The data presented in this study are available upon request from the corresponding author. The data are not publicly available due to patient privacy and data restrictions.

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
