# Peer review of "Comparative Evaluation of a Standard M10 Assay with Xpert Xpress for the Rapid Molecular Diagnosis of SARS-CoV-2, Influenza A/B Virus, and Respiratory Syncytial Virus"

_diagnostics, 2023, doi:10.3390/diagnostics13233507_

Round 1

Reviewer 1 Report

Comments and Suggestions for Authors

The authors submit a well-presented comparative study identifying performance of the Standard M10 test for IAV, IBV, RSV and SARS-CoV-2. Data are rationale and support the conclusions.

Reviewer 2 Report

Comments and Suggestions for Authors

Very well written article, which confirms, in another biotope, the results previously reported by an Italian team in 2023 on the STANDARD M10 Flu/RSV/SARS-CoV-2 assay from SD Biosensor Inc. (Domnich et al. J Clin Virol 2023;161:105402). These results are important for tropical countries, where SARS-CoV-2, the influenza viruses and RSV circulate throughout the year. It is however possible to improve the manuscript.

1°. Please express all Ct values in arbitrary units (a.u.);

2°. Figure 2: Abscissa line: Better to plot XpertXpress and STANDARD M10 on the same line (rather than the current offset);

3°. Figure 2: Indicate the genes of Ct for the different viruses;

4°. Figure 2: Complete the Figure 2 with Bland-Alaman analyses, including description in the Materials and Methods section and Bland-Altman results in the Results section;

5°. Line 231: Delete "Table 1";

6°. Lines 248 and 249: Add a.u. after the Ct values;

7°. Add a full Limitations of the study paragraph at the end of the Discussion section (retrospective study; conservation of samples; limited sizes of samples for the study viruses; etc);

8°. Add or discuss a usuability study including satisfaction study; discuss the duration of the test for one sample; the number of samples per hour; Discuss the workflow (in comparison with XpertXpress);

9°. Introduce elements on the cost of the test (analyzer; cost of reagents; internal quality controls; etc.);

10°. Finally, discuss in which contexts (emergencies; developing countries; etc, etc.) this new test could be proposed and used in practice.

Reviewer 3 Report

Comments and Suggestions for Authors

This work presents a fairly extensive comparison of several commercial diagnostic systems based on their effectiveness in identifying pathogens of respiratory diseases. The main problem of this study is its novelty, since the described Standard M10 diagnostic system is already well described (https://doi.org/10.1016/j.jcv.2023.105402). The authors should clarify what makes this study unique.

The conclusion about the suitability of this system for point-of-care diagnostics is also not clear. This does not follow from the results presented.

minor:

Materials and Methods states that 150 negative samples were collected, but other sections describe only 107 - please explain.

Reviewer 4 Report

Comments and Suggestions for Authors

Abdullah et al. have reported on an evaluation of a commercial real-time PCR assay for respiratory viruses. Before publication is considered, I have the following suggestions on how the manuscript can be further improved.

1.) As stated by the authors, they have performed their test comparison in line with STARD recommendations. However, as STARD recommendations the presentation at least of basic characteristics of the study population, respective data should be added.

2.) The authors should more thoroughly explain the choice of their sample count, otherwise, their decision may make an arbitrary impression.

3.) Minor point, discussion second sentence: The term “gold standard” is poorly chosen, because a true gold standard is defined by (hypothetical) perfect diagnostic accuracy, i.e. sensitivity and specificity of 100% each. As virus culture rarely shows 100 sensitivity but is vulnerable to die-off phenomena, it can be hardly considered as a gold standard.

4.) Title and text: It seems more appropriate to speak of a “test comparison” rather than of a “test evaluation”, because relevant features of an evaluation like repeated assessment of inter- and intra-assay reproducibility of obtained results are not presented.

Round 2

Reviewer 3 Report

Comments and Suggestions for Authors

The authors answered my questions and significantly improved the manuscript. The manuscript can now be published in the "Diagnostics" journal.